# GNN-based Probabilistic Supply and Inventory Predictions in Supply Chain Networks

## Abstract

Successful supply chain optimization must mitigate imbalances between supply and demand over time. While accurate demand prediction is essential for supply planning, it alone does not suffice. The key to successful supply planning for optimal and viable execution lies in maximizing predictability for both demand and supply throughout an execution horizon. Therefore, enhancing the accuracy of supply predictions is imperative to create an attainable supply plan that matches demand without overstocking or understocking. However, in complex supply chain networks with numerous nodes and edges, accurate supply predictions are challenging due to dynamic node interactions, cascading supply delays, resource availability, production and logistic capabilities. Consequently, supply executions often deviate from their initial plans. To address this, we present the Graph-based Supply Prediction (GSP) probabilistic model. Our attention-based graph neural network (GNN) model predicts supplies, inventory, and imbalances using graph-structured historical data, demand forecasting, and original supply plan inputs. The experiments, conducted using historical data from a global consumer goods company's large-scale supply chain, demonstrate that GSP significantly improves supply and inventory prediction accuracy, potentially offering supply plan corrections to optimize executions.

## 1 Introduction

At the heart of supply chain optimization lies the task of mitigating the risks associated with imbalances between supply and demand across a supply chain network over time (Kleindorfer & Saad (2005); Tang & Tomlin (2008); Ivanov (2022); Abdel-Basset et al. (2019)). While accurate demand prediction is a critical factor in optimizing supply planning, it alone does not suffice (Manuj & Mentzer (2008)). Successful supply planning in the context of optimal and viable execution hinges on the ability to achieve the highest degree of predictability for both demand and supply, spanning the execution horizon. Making better supply and inventory predictions is imperative to create an attainable supply plan that effectively matches demand without overstocking or understocking inventory. While considerable efforts have been directed toward improving demand prediction in isolation (Salinas et al. (2020); Makridakis et al. (2022); Hyndman & Athanasopoulos (2018)), comparatively less attention has been given to predicting supply events (quantity/timing), lead times, and inventory levels as an integrated whole, accounting for all variabilities in demand and supply.

However, in complex, large-scale supply chain networks with a multitude of nodes and edges (Figure 1), making accurate supply and inventory predictions across all nodes and edges poses significant challenges. This difficulty arises from the need to account for dynamic interactions among interconnected nodes in the network, the ripple effects of supply delays through multi-hop nodes, resource availability, production capacity, and logistics capabilities.

Typically, planned shipments from the Sales and Operations Planning (S&OP) process, which consider limited sets of states, conditions, and constraints, tend to be notably inaccurate and unsuitable for direct execution. Consequently, organizations need to bridge the gap between S&OP shipment plans and day-to-day operational shipment activities, aligning them with financial objectives and tackling supply and demand challenges (Hippold (2019); Hainey (2022)). Therefore, supply chain operators frequently encounter situations where they cannot act on the planned shipments due to stock shortages and delayed upstream supplies. Accurate predictions of the actual executed events

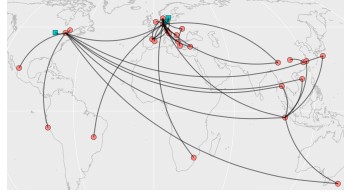

Figure 1: Supply Chain Network (Example)

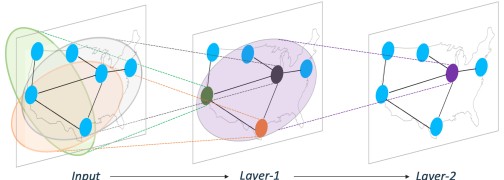

Figure 2: Graph Node Embedding

(quantity/timing) for planned shipments support operators in achieving better alignment with day-to-day operational objectives.

To address the shipment event, supply and inventory prediction problems, this paper introduces the Graph-based Supply Prediction (GSP) probabilistic model which is tailored for situations where planned shipment and forecasted demand inputs are available over the specified time horizon. We employ attention-based graph neural networks (GNN) to make network-wide consistent and simultaneous predictions for incoming and outgoing supplies and inventory, relying on sequential graph-structured snapshots of historical supply chain data, demand forecasting, and shipment plan inputs.

In addition, we propose a model-training loss function that combines cumulative supply prediction errors with inventory prediction errors. To elaborate further, incorporating cumulative supply prediction errors as illustrated in Figure 3 into the loss function deals with the prediction inaccuracies attributed to the unpredictable supply variability in both the quantity and timing of shipment events throughout the time horizon.[1] Note that in an edge with a consistent lead time, the cumulative outgoing supply quantity from the source node over the time horizon has a proportional impact on the inventory level at the destination node.

Moreover, it is worth noting that frequently, our primary objective extends beyond merely predicting the supply itself. Instead, we often focus on forecasting key performance metrics such as service level, fill rate, and the total economic cost associated with imbalanced risks (e.g., lost sales and excess inventory). In this context, it is vital to train the model to predict both inventory levels and supply coherently, taking into account the provided demand prediction inputs. The inclusion of inventory prediction errors in the loss function accounts for comprehensive impacts of demand and supply variabilities on the accuracy of inventory predictions, since the inventory of each node is a result of the cumulative sums of incoming supply, outgoing supply, and demand.

The experiments, conducted using historical data from a complex supply chain network of a global consumer goods company, demonstrate that GSP achieves substantial enhancements in both supply and inventory prediction accuracy. These improvements have the potential to drive corrective adjustments in supply plans, ultimately leading to more optimal executions.

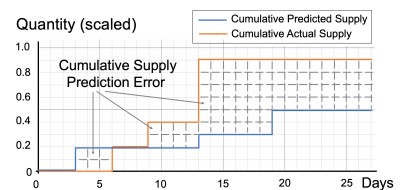

Figure 3: Cumulative Supply Prediction Error

In summary, our contributions can be highlighted in three aspects.

---

[1]Consider a scenario with actual quantities [0, 100, 0, 0] over 4 timesteps. Evaluating three predictive models, such as M1 = [0, 0, 100, 0], M2 = [100, 0, 0, 0], and M3 = [0, 0, 0, 0], M1 and M2 have a wMAPE of 200%, while M3 has 100%. We emphasize that wMAPE (= weighted Mean Absolute Percentage Error = the sum of the absolute quantity errors (= samples of |predicted quantity - actual quantity|) divided by the sum of the actual quantities) isn't suitable for event prediction evaluation, since it doesn't account for both quantity and timing predictions in a single metric. Thus, we introduce **sMACE (scaled Mean Absolute Cumulative Error)**, defined as the mean of the absolute cumulative quantity errors (= samples of |cumulative predicted quantity - cumulative actual quantity|) divided by the mean of the actual quantities. Refer to Appendix C for a detailed discussion on sMACE. In the context of sMACE, when an actual event quantity (representing a step increase in the cumulative actual quantity function in Figure 3) is predicted to occur either $d$ timesteps earlier ($d < 0$) or later ($d > 0$), it contributes an error equal to the quantity multiplied by $|d|$. In terms of sMACE, both M1 and M2 have a score of 100%, whereas M3 scores 300%.

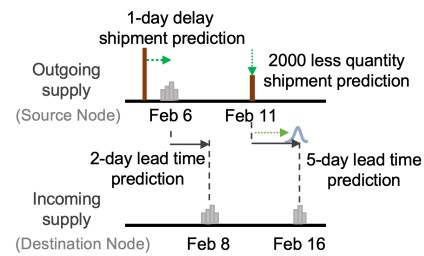

Figure 4: Edge-level Shipments (Example)

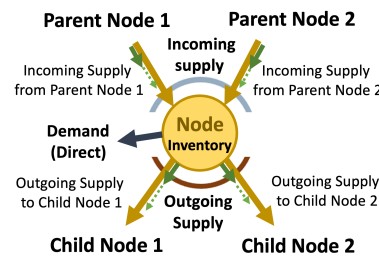

Figure 5: Node-level Relationship (Example)

- We presented a novel GNN-based generalized method for predicting event quantity/timing in graph-structured problem contexts where there are planned events without a one-to-one mapping to actual events. This implies that there is no ground truth as labeled data for quantity/timing variables for model learning. Our approach relies on labeled data in the form of edge-level and/or node-level aggregated quantities at specific temporal granularities, such as daily or weekly periods.

- We proposed a novel prediction error metric called sMACE (scaled Mean Absolute Cumulative Error, detailed in Appendix C) to assess the inaccuracies in predictions caused by unpredictable supply variability in both quantity and timing of events. Also, we employed the concept of sMACE in the loss function for training our GNN-based event delta prediction models.

- In the context of supply chain networks, we built GSP models using the GNN-based generalized method, specifically developed for supply chain network scenarios aiming to predict outgoing shipment events (quantity/timing). We demonstrated network-wide, reliable supply and inventory predictions for supply and inventory while adhering to node-level supply capacity constraints, conducting experiments with a global consumer goods company's large-scale real supply chain data.

The structure of this paper is as follows: In Section 2, we conduct a comparison with related work. Section 3 describes our problem, followed by the presentation of our generalized method in Section 4. Section 5 introduces the GSP models, while Section 6 showcases our experiment results. The paper concludes with Section 7.

## 2 RELATED WORK

Predicting supply events requires unique techniques, distinct from individual node-level event prediction methods for intermittent demand forecasting (Kourentzes (2014); Petropoulos & Kourentzes (2015)), such as Croston's method and its modifications (Croston (1972); Syntetos & Boylan (2005); Teunter & Sani (2009)), and deep renewal process (Turkmen et al. (2019)). When dealing with supply event predictions across an entire network, it is crucial to account for the cascading effect of events as they traverse various network nodes and edges with distinct topological structures. Our approach implements a GNN-based iterative inference technique to maintain prediction consistency at all node and edge levels across the entire network.

Our problem differs significantly from traditional lead time prediction or ETA (estimated time of arrival) prediction scenarios (Viellechner & Spinler (2020); Mariappan et al. (2023); Hathikal et al. (2020); Viellechner & Spinler (2020); Lingitz et al. (2018); Gyulai et al. (2018)). In those contexts, the target variable is typically defined as the time duration between the planned order initiation and its eventual arrival at the destination. In our GSP probabilistic approach, we broaden the scope to encompass detailed predictions. Our primary focus lies in forecasting both the timing and quantity of shipments originating from the source node. Subsequently, we integrate these shipment predictions with the lead time predictions between the outgoing shipment and the reception event. These comprehensive predictions from both source and destination nodes' viewpoints (Figure 4) play a pivotal role in ensuring coherent and causally explainable reconstruction of node-level inventory predictions (Figure 5), drawing upon edge-level shipment quantity/timing predictions and node-level demand forecasts.

## 3 PROBLEM DESCRIPTION

We consider supply chain networks where each SKU is associated with its own distinct topological graph. Let $\mathcal{G} = (\mathcal{V}, \mathcal{E})$ denote a directional graph for the SKU-specific supply chain network [2], containing a collection of nodes $\mathcal{V} = \{1, ..., n\}$ with diverse node types (e.g., plants, distribution centers, retailers) and edges $\mathcal{E} \in \{\mathcal{V} \times \mathcal{V}\}$ where $(v, w) \in \mathcal{E}$ denotes an edge from a source node $v$ to a destination node $w$. The nodes and edges in $\mathcal{G}$ at time $t$ are associated with a set of node feature vectors $\mathbf{x}^t = \{\mathbf{x}_v^t \in \mathbb{R}^{d_1} : v \in \mathcal{V}\} \in \mathbb{R}^{|\mathcal{V}| \times d_1}$ and a set of edge feature vectors $\mathbf{a}^t = \{\mathbf{a}_{vw}^t \in \mathbb{R}^{d_2} : (v, w) \in \mathcal{E}\} \in \mathbb{R}^{|\mathcal{E}| \times d_2}$ where $d_1$ is the number of node features and $d_2$ is the number of edge features. In addition, let $\mathcal{G}^R = (\mathcal{V}, \mathcal{E}^R)$ denote the reverse graph of $\mathcal{G}$ with the same nodes and features but with the directions of all edges reversed, that is, $\mathcal{E}^R = \{(w, v) | (v, w) \in \mathcal{E}\}$.

Our models make supply and inventory predictions in supply chain networks where the input data on planned (scheduled) shipment events, weekly demand forecasting and other planning features over the specified time horizon are available. Note that the planned shipment (quantity/timing) events are the original supply plan acquired from the enterprise's planning system, which frequently falls short of feasible for execution as originally planned.

The objective is to minimize both (a) the average absolute errors of edge-level daily outgoing supply cumulative predictions and (b) the average absolute errors of node-level weekly inventory predictions. More precisely, our evaluation metrics are defined in a normalized manner as follows.

(a) daily outgoing supply prediction sMACE

$$\mathbf{sMACE} = \frac{\sum_{(t,(v,w)) \sim \mathcal{D}} \sum_{h \in H} |\hat{Q}_{vw}^{\mathrm{day},t}(h) - Q_{vw}^{\mathrm{day},t}(h)|}{\sum_{(t,(v,w)) \sim \mathcal{D}} \sum_{h \in H} q_{vw}^{\mathrm{day},t}(h)} \times 100\ \% \qquad (1)$$

where the predicted and actual cumulative daily quantity vectors of outgoing supply are defined as:

$$\hat{\mathbf{Q}}_{vw}^{\mathrm{day},t} = \{\hat{Q}_{vw}^{\mathrm{day},t}(h) = \sum_{d=0}^{h} \hat{q}_{vw}^{\mathrm{day},t}(d) \mid h \in H\}, \quad \mathbf{Q}_{vw}^{\mathrm{day},t} = \{Q_{vw}^{\mathrm{day},t}(h) = \sum_{d=0}^{h} q_{vw}^{\mathrm{day},t}(d) \mid h \in H\} \quad (2)$$

Here, $\hat{q}_{vw}^{\mathrm{day},t}(d)$ is the predicted daily quantity of outgoing supply for day $d$ in a time horizon ($|H|$ days, $H = \{0, 1, 2, ..., |H| - 1\}$) from source node $v$ to destination node $w$ at the prediction time $t$. Also, $q_{vw}^{\mathrm{day},t}(d)$ is the actual daily quantity (as ground truth) of outgoing supply for day $t + d$ on the edge from $v$ to $w$.

(b) weekly inventory prediction wMAPE

$$\mathbf{wMAPE} = \frac{\sum_{(t,v) \sim \mathcal{D}} \sum_{\mathrm{w} \in W} |\hat{I}_v^{\mathrm{week},t}(\mathrm{w}) - I_v^{\mathrm{week},t}(\mathrm{w})|}{\sum_{(t,v) \sim \mathcal{D}} \sum_{\mathrm{w} \in W} I_v^{\mathrm{week},t}(\mathrm{w})} \times 100\ \% \qquad (3)$$

where $\hat{I}_v^{\mathrm{week},t}(\mathrm{w})$ and $I_v^{\mathrm{week},t}(\mathrm{w})$, respectively, are the weekly predicted and actual inventory for node $v$ over the weekly time horizon $W = \{0, 1, 2, ..., |W| - 1\}$ at week $t$. Here, $|W| = |H|/7$.

## 4 GENERALIZED METHOD FOR EVENT DELTA PREDICTIONS

Before introducing GSP models for shipment event predictions in supply chain networks, we describe the generalized method for GNN-based event quantity/timing delta predictions in graph-structured problem contexts where there is information about planned events, but there is no one-to-one mapping with actual events for these planned events. Therefore, in our problem setting, there is no ground truth available as labeled data for quantity/timing variables in model learning. However, we assume that labeled data (ground truth) is obtainable in the form of edge-level and/or node-level aggregated quantities at specific temporal granularities, such as in daily or weekly periods.

---

[2]For the ease of notation and without loss of generality, we drop the SKU-specific subscript from most notations, unless we explicitly include it.

## 4.1 Graph Attention Networks

GNNs are highly effective in representing graph-structured data by generating graph node embeddings (Bronstein et al. (2021); Hamilton et al. (2017); Kipf & Welling (2017)). These embeddings are created through graph convolutions (Figure 2) that consider both node-level and edge-level features, making them particularly well-suited for capturing the complex dynamics of demand and supply interactions among interconnected nodes via edges.

To achieve this, we harness the power of Graph Attention Networks (GAT), a specific type of GNN equipped with dynamic attention mechanisms that empower our model with the capability to dynamically allocate different weights to diverse connections, drawing on information from the present states of both nodes and edges (Brody et al. (2022); Veličković et al. (2018)).

In GAT, dynamic attentions enable a node to selectively attend to its neighboring nodes that are highly relevant in its supply predictions. Provided both node features $\mathbf{h}^{(0)}(= \mathbf{x})$ and edge features $\mathbf{e}\ (= \mathbf{a})$ for $\mathcal{G} = (\mathcal{V}, \mathcal{E})$, $\mathrm{GAT}_{\mathrm{XA}}$ (Appendix A) denotes the $L$-layered graph convolution network that makes iterative updates $l = 1, 2, ..., L$ and calculates

$$\mathbf{h}^{(L)} = \mathrm{GAT}_{\mathrm{XA}}(\mathbf{h}^{(0)}, \mathbf{e}, \mathcal{G}; \phi_{\mathrm{XA}}) = \{\mathrm{h}_v^{(L)} \in \mathbb{R}^B : v \in \mathcal{V}\} \in \mathbb{R}^{B \times |\mathcal{V}|} \qquad (4)$$

where $\phi_{\mathrm{XA}}$ is the set of all learned parameters of $\mathrm{GAT}_{\mathrm{XA}}$ and $B$ is the embedding dimension. The graph node embedding $\mathbf{u}$ for a given pair of node features $\mathbf{x}$ and edge features $\mathbf{a}$ is defined as:

$$\mathbf{u} \overset{\text{def}}{=} \mathrm{emb}_{\mathrm{XA}}(\mathbf{x}, \mathbf{a}; \phi_{\mathrm{XA}_f}, \phi_{\mathrm{XA}_b}) = [\mathbf{u}_f \parallel \mathbf{u}_b] \in \mathbb{R}^{2B \times |\mathcal{V}|} \qquad (5)$$

where $\mathbf{u}_f = \mathrm{GAT}_{\mathrm{XA}}(\mathbf{x}, \mathbf{a}, \mathcal{G}; \phi_{\mathrm{XA}_f})$ and $\mathbf{u}_b = \mathrm{GAT}_{\mathrm{XA}}(\mathbf{x}, \mathbf{a}, \mathcal{G}^R; \phi_{\mathrm{XA}_b})$. Note that $\mathbf{u}_f$ and $\mathbf{u}_b$ each are calculated using the forward-directional (original) graph and the backward-directional (reverse) graph, respectively. The $\mathbf{u}_f$ embedding vectors from graph $\mathcal{G}$ incorporate the competing and collaborative dynamics of multiple source nodes to a destination node into their own attentions, whereas the $\mathbf{u}_b$ embedding vectors from reversed graph $\mathcal{G}^R$ encompass the interactions of multiple destination nodes to a source node into their respective attentions.

## 4.2 Event Quantity/Timing Delta Prediction for Each Planned Event

In this section, we present the probabilistic event prediction model that leverages GAT graph embedding to predict the timings and quantities of events at every edge, provided the information on the timings and quantities of originally planned events.

Specifically, let $\tau_{vw}^{i|t} \in H = \{0, 1, 2, ..., |H| - 1\}$ be the originally planned time of event $i$ elapsed from the current time $t$. That is, the event $i$ on the edge from $v$ to $w$ is planned to occur at time $t + \tau_{vw}^{i|t}$. Also, we denote the planned outgoing shipment quantity of event $i$ by $a_{vw}^{i|t} \in \mathbb{R}$.

Given $\tau_{vw}^{i|t}$ and $a_{vw}^{i|t}$ of the $i$-th planned event ($i \in \{1, 2, .., |H|\})$[3] on the edge from $v$ to $w$ at the prediction time $t$, we predict $P_{vw}^{i|t}(\boldsymbol{\delta})$ and $\hat{a}_{vw}^{i|t}$. $P_{vw}^{i|t}(\boldsymbol{\delta})$ is the predicted discrete probability distribution of the time difference variable $\boldsymbol{\delta}$ (in days) between the actual event time and the originally planned event time where $\boldsymbol{\delta} \in \Delta = \{-7, -6, ..., -1, 0, 1, ..., 6, 7\}$, and $p_{vw}^{i|t}(\delta)$ is the probability of predicted time difference $\boldsymbol{\delta} = \delta$. Also, the predicted shipment quantity $\hat{a}_{vw}^{i|t} = r_{vw}^{i|t} a_{vw}^{i|t} \in \mathbb{R}$ is calculated using $r_{vw}^{i|t} \in (0, 2]$ as a predicted multiplier.[4] Figure 6 illustrates the comparisons among predicted, planned, and actual shipments. We denote

$$\mathbf{r}^{i|t} = \{r_{vw}^{i|t} \mid (v, w) \in \mathcal{E}\} \in \mathbb{R}^{|\mathcal{E}|}; \ \ \mathbf{P}^{i|t}(\boldsymbol{\delta}) = \{P_{vw}^{i|t}(\boldsymbol{\delta}) \mid (v, w) \in \mathcal{E}\} \in \mathbb{R}^{|\mathcal{E}| \times |\Delta|}. \qquad (6)$$

---

[3]The predictions of $|H|$ events cover the maximum possible count of daily events over the $|H|$ days.

[4]Alternatively, the predicted shipment quantity may be modeled to depend on $\delta$: $\hat{a}_{vw}^{i|t}(\delta) = \min\{0, (r_{vw}^{i|t} + \delta s_{vw}^{i|t})\} a_{vw}^{i|t} \in \mathbb{R}$ where $r_{vw}^{i|t} \in (0, 2]$ and $s_{vw}^{i|t} \in [-1, 1]$ are predicted multipliers. Note that $s_{vw}^{i|t}$ may capture a potential correlation between the time difference (earlier or later than the planned shipment time) and the shipment quantity (larger or smaller than the quantity on time). Furthermore, $P(\hat{a}_{vw}^{i|t}, \delta_{vw}^{i|t}) = P(\hat{a}_{vw}^{i|t} \mid \delta_{vw}^{i|t}) P(\delta_{vw}^{i|t})$. An extension employing Bayesian regression allows for modeling the quantity prediction as a conditional probability $P(\hat{a}_{vw}^{i|t} \mid \delta_{vw}^{i|t})$.

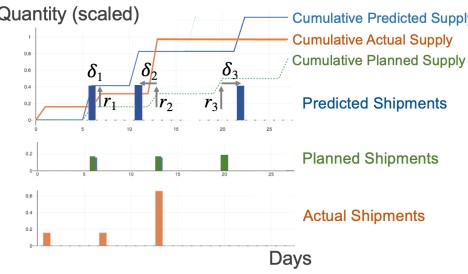

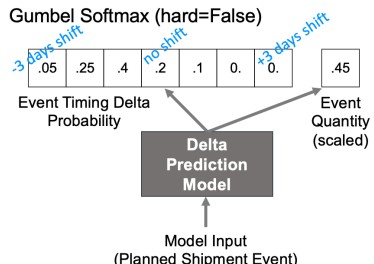

Figure 6: Predicted/Planned/Actual Shipments    Figure 7: Event Timing/Quantity Prediction

For each $i$, the GAT-based prediction model $M$ predicts $i$-th event's

$$(\mathbf{r}^{i|t}, \mathbf{P}^{i|t}(\boldsymbol{\delta})) = M(\mathbf{x}^t, \mathbf{a}^t(i)) \tag{7}$$

at time $t$, taking edge-level features $\mathbf{a}^t(i) = \{\mathrm{a}^t_{vw}(i) \mid (v,w) \in \mathcal{E}\}$ and node-level features $\mathbf{x}^t = \{\mathrm{x}^t_v \mid v \in \mathcal{V}\}$ as inputs. For simplicity, we assume that node-level features are dependent only on the prediction time $t$, not the prediction target event $i$. For the edge-level features, as the default setting, we use

$$\mathrm{a}^t_{vw}(i) = \{\tau^{i|t}_{vw}, a^{i|t}_{vw}\} \cup \{\tau^{-k|t}_{vw,\mathrm{hist}}, a^{-k|t}_{vw,\mathrm{hist}} \mid k = 0, 1, 2, ..., K-1\} \tag{8}$$

where $\tau^{-k|t}_{vw,\mathrm{hist}}$ and $a^{-k|t}_{vw,\mathrm{hist}}$ represents historical shipment event times and quantities, with $k = 0$ denoting the most recent event before time $t$.

To predict $\mathbf{r}^{i|t}$ and $\mathbf{P}^{i|t}(\boldsymbol{\delta})$, the model $M$ begins by calculating GAT embeddings $\mathbf{u}_f$ and $\mathbf{u}_b$, taking $\mathbf{x}^t$ and $\mathbf{a}^t(i)$, as outlined in Equation 5. Then, using the graph node embeddings $u_v = [u_{f,v} \parallel u_{b,v}]$ and $u_w = [u_{f,w} \parallel u_{b,w}]$ for nodes $v$ and $w$,

$$r^{i|t}_{vw} = \mathtt{mlp}_r([u_v \parallel u_w]) \quad \in (0,2) \tag{9}$$

where $\mathtt{mlp}_r$ is a multilayer feedforward network with an output of $\mathtt{sigmoid}$ multiplied by 2.0. Also,

$$P^{i|t}_{vw}(\boldsymbol{\delta}) = \mathtt{GumbelSoftmax}[\mathtt{mlp}_p([u_v \parallel u_w])] \quad \in [0,1]^{|\Delta|} \tag{10}$$

where $\mathtt{mlp}_p$ is a multilayer feedforward network and $\mathtt{GumbelSoftmax}$ (Figure 7) is to sample from a categorical distribution in the forward pass and be differentiable in backprop (Jang et al. (2016)). Note that the predicted time of an event $\tau^{i|t}_{vw} + \delta$ should be always zero or a positive integer. Thus, we only allow $\delta \geq -\tau^{i|t}_{vw}$, and update the probability of $\delta = 0$ by $p^{i|t}_{vw}(0) := \sum_{\delta' < -\tau^{i|t}_{vw}} p^{i|t}_{vw}(\delta')$.

### 4.3 Predicted Event Aggregation into Edge-Level Quantities over Time

In this paper, we exhibit the aggregation of the event predictions into the vector of daily event quantities over a defined time horizon ($|H|$ days, $H = \{0, 1, 2, ..., |H| - 1\}$), denoted as $\hat{q}^{\mathrm{day},t}_{vw} \in \mathbb{R}^{|H|}$ on the edge from node $v$ to node $w$ at the prediction time $t$ (= the start of day $t$).[5] However, our methodology can be flexibly applied to time granularities other than the daily level.

For any event time $t' \in H$ elapsed from the current time $t$, $e^{(t')} \in \mathbb{R}^{|H|}$ is the standard basis vector $[0, \ldots, 0, 1, 0, \ldots, 0]$ with a 1 at position $t' + 1$.[6] It is worth mentioning that the probability distribution of the planned event time is described as $e^{(\tau^{i|t}_{vw})} \in \mathbb{R}^{|H|}$, which assigns the full probability 1 to the element corresponding to the planned event time $\tau^{i|t}_{vw}$.

Using $p^{i|t}_{vw}(\delta)$ and $\tau^{i|t}_{vw}$, we calculate the probability distribution of the predicted event time $\hat{\tau}^{i|t}_{vw}$ over the time horizon $H$ by

$$\boldsymbol{\pi}^{i|t}_{vw} = \sum_{\delta \in \Delta} p^{i|t}_{vw}(\delta) \, e^{(\tau^{i|t}_{vw} + \delta)} \in \mathbb{R}^{|H|}. \tag{11}$$

---

[5]At the prediction time $t$ (= the start of day $t$), we designate the prediction for the same day $t$ as the *forecasted timestep 0* (or $h = 0$) prediction. As a result, the prediction made for the final day of the $|H|$ day horizon is referred to as the *forecasted timestep* $|H| - 1$ (or $h = |H| - 1$) prediction.

[6]Since the minimum value of $t'$ is zero, the first element of vector $e^{(t')}$ corresponds to when $t' = 0$. Also, for ease of performing mathematical operations below, we set $e^{(t')}$ to the zero vector $\mathbf{0}$ if $t' \notin H$.

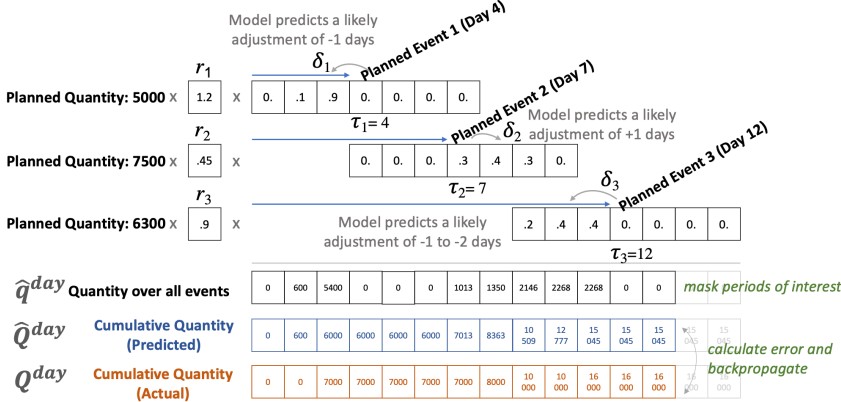

Figure 8: An Illustration of $H = 14$ Day Prediction with Planned Shipments on Days 4, 7, 12

Then, the predicted quantity distribution vector of event $i$ over the time horizon $H$ is

$$\hat{\boldsymbol{q}}_{vw}^{i|t} \;=\; r_{vw}^{i|t} a_{vw}^{i|t} \boldsymbol{\pi}_{vw}^{i|t} \;=\; r_{vw}^{i|t} a_{vw}^{i|t} \sum_{\delta \in \Delta} p_{vw}^{i|t}(\delta)\, \boldsymbol{e}^{(\tau_{vw}^{i|t}+\delta)} \in \mathbb{R}^{|H|}. \tag{12}$$

The predicted daily quantity vector is computed as:

$$\hat{\boldsymbol{q}}_{vw}^{\mathrm{day},t} = \sum_{i \in \mathcal{A}_{vw}^{t}} \hat{\boldsymbol{q}}_{vw}^{i|t} \in \mathbb{R}^{|H|} \tag{13}$$

where $\mathcal{A}_{vw}^{t} = \{i \in \mathbb{N} \mid \tau_{vw}^{i|t} \in H\}$ is the set of all planned events at time $t$ over the time horizon $H$.

The predicted and actual cumulative daily quantity vectors are defined as:

$$\hat{\mathbf{Q}}_{vw}^{\mathrm{day},t} = \{\hat{Q}_{vw}^{\mathrm{day},t}(h) = \sum_{d=0}^{h} \hat{q}_{vw}^{\mathrm{day},t}(d) \mid h \in H\}, \quad \mathbf{Q}_{vw}^{\mathrm{day},t} = \{Q_{vw}^{\mathrm{day},t}(h) = \sum_{d=0}^{h} q_{vw}^{\mathrm{day},t}(d) \mid h \in H\} \tag{14}$$

Note that $\hat{q}_{vw}^{\mathrm{day},t}(d)$ is the predicted daily quantity for day $d$ in $H$ on the edge from $v$ to $w$ at the prediction time $t$. Also, $q_{vw}^{\mathrm{day},t}(d)$ is the actual daily quantity (as ground truth) for day $t + d$ on the edge from $v$ to $w$.

The set of parameters of the prediction model $\theta_M = \{\phi_{\mathrm{XA_f}}, \phi_{\mathrm{XA_b}}, \phi_{\mathrm{mlp}_p}, \phi_{\mathrm{mlp}_r}\}$. Figure 8 illustrates the calculation of predicted vectors over $H = 14$ days for given $t$ and $(v, w)$.

### 4.4 PREDICTED EVENT AGGREGATION INTO NODE-LEVEL QUANTITIES OVER TIME

Suppose that there are node-level labeled quantities (as ground truth) that we aim to incorporate into our model training for event quantity/timing delta predictions, along with the edge-level labeled quantities. Also, we assume that there exists a pre-determined process model $Z$ that takes edge-level predicted quantity vectors $\{\hat{\boldsymbol{q}}_{vw}^{\mathrm{day},t} \mid (v, w) \in \mathcal{E}\}$ as inputs and predicts node-level aggregated quantity vectors $\{\hat{\mathbf{I}}_{v}^{\mathrm{week},t} \mid v \in \mathcal{G}\}$ in a different time granularity (week for an illustration here) as outputs where $\hat{\mathbf{I}}_{v}^{\mathrm{week},t} = \{\hat{I}_{v}^{\mathrm{w}|t} \mid \mathrm{w} \in W\}$ for the weekly time horizon $W = \{0, 1, 2, ..., |W| - 1\}$ and $|W| = |H|/7$. That is,

$$\{\hat{\mathbf{I}}_{v}^{\mathrm{week},t}\} = Z(\{\hat{\boldsymbol{q}}_{vw}^{\mathrm{day},t}\}) \tag{15}$$

We notate node-level labeled quantities (as ground truth) by $\mathbf{I}_{v}^{\mathrm{week},t} = \{I_{v,\mathrm{actual}}^{t+\mathrm{w}} \mid \mathrm{w} \in W\}$ where $I_{v,\mathrm{actual}}^{t+w}$ is the actual quantity in node $v$ at the start of time $t$.

### 4.5 LOSS FUNCTION

Let $\theta_M$ be the set of parameters of prediction model $M(\mathbf{x}, \mathbf{a} ; \theta_M)$. The loss function combines edge-level cumulative outgoing supply prediction errors with node-level inventory prediction errors.

$$\mathcal{L}(\theta_M) = (1-\alpha)\, \mathbb{E}_{(t,(v,w))\sim\mathcal{D}}[\ \|\ \hat{\mathbf{Q}}_{vw}^{\mathrm{day},t} - \mathbf{Q}_{vw}^{\mathrm{day},t}\ \|_2^2\ ] + \alpha\, \mathbb{E}_{(t,v)\sim\mathcal{D}}[\ \|\ \hat{\mathbf{I}}_{v}^{\mathrm{week},t} - \mathbf{I}_{v}^{\mathrm{week},t}\ \|_2^2\ ] \tag{16}$$

where $\alpha \in [0, 1]$ is a hyperparameter that controls the balance of the two loss components. When the loss function is set with $\alpha = 0$, it enables model training exclusively with edge-level labeled quantities. Conversely, when $\alpha = 1$, it does model training solely with node-level labeled quantities. We optimize $\theta_M$ by minimizing the loss $\mathcal{L}(\theta_M)$ over the dataset $\mathcal{D}$ through backprop.

## 5 GRAPH-BASED SUPPLY PREDICTION (GSP) MODELS

We build the GSP models by applying the generalized method outlined in Section 4 to supply chain network scenarios where the goal is to predict outgoing shipment events (quantity/timing). Due to limitations in the underlying supply chain processes and systems for tracking actual shipments against planned shipments, it is often not possible to establish a direct one-to-one correspondence between actual and planned shipments. Consequently, we lack precise information regarding the exact discrepancies between actual and planned shipments in terms of both shipment event quantity and timing. Nevertheless, there are the edge-level daily outgoing supply quantities ($\{q_{vw}^{\mathrm{day},t}\}$) and the node-level weekly inventory quantities ($\{\mathbf{I}_v^{\mathrm{week},t}\}$) available for use as ground-truth labels in model training and evaluation. The prediction model $M$ in Equation 7 takes as inputs the edge-level features, configured by default in Equation 8, along with the node-level features defined as:

$$
\begin{aligned}
\mathrm{x}_v^t = \{I_{v,\mathrm{actual}}^t\} \;\; &\cup \;\; \{I_{v,\mathrm{plan}}^{\mathrm{w}|t} \,|\mathrm{w} = 1, 2, ..., |W| - 1\} \\
&\cup \; \{D_{v,\mathrm{pred}}^{\mathrm{w}|t}, S_{v,\mathrm{plan}}^{\mathrm{w}|t}, A_{v,\mathrm{plan}}^{\mathrm{w}|t} \,|\mathrm{w} = 0, 1, ..., |W| - 1\}.
\end{aligned}
\tag{17}
$$

$I_{v,\mathrm{actual}}^t$ is the actual inventory at the start of time $t$. Also, $I_{v,\mathrm{plan}}^{\mathrm{w}|t}$, $D_{v,\mathrm{pred}}^{\mathrm{w}|t}$, $S_{v,\mathrm{plan}}^{\mathrm{w}|t}$, and $A_{v,\mathrm{plan}}^{\mathrm{w}|t}$ are planned inventory, predicted demand, planned incoming supply, and planned outgoing supply for week w as predicted at the start of week $t$, respectively. In this paper we suppose that these weekly demand forecasting and planning features are obtained from the organization's existing system of planning.

We compute the node-level weekly predicted inventory vectors $\{\hat{\mathbf{I}}_v^{\mathrm{week},t}\} = Z(\{\hat{q}_{vw}^{\mathrm{day},t}\})$ using the inventory prediction process model $Z$, described in Appendix D. Note that the model $Z$ internally relies on an edge-level model for probabilistic discrete lead time prediction $P_{\mathrm{LT},vw}^{t+h}(\mathrm{k})$ where $p_{\mathrm{LT},vw}^{t+h}(k)$ is the probability of predicted lead time $\mathrm{k} = k \in H$ for any outgoing supply at day $t + h$.

In the context of network-wide supply event predictions, it becomes essential to predict the sequence of events propagating through nodes and edges. This requires connected predictions that span across nodes and edges in the network. Appendix F illustrates our approach for iterative and simultaneous predictions that allow for satisfying each node's supply capacity constraint, which is affected by the supply executions of other neighboring nodes in a cascading manner.

## 6 EXPERIMENT RESULTS

| Method | (a) Daily Outgoing Supply sMACE | (b) Weekly Inventory wMAPE | (c) Weekly Constraint Error $\kappa$ |
|---|---|---|---|
| GSP ($\alpha$=0.0) | $102.90 \pm 0.1$ % | $31.03 \pm 0.3$ % | $3.99 \pm 0.03$ % |
| GSP ($\alpha$=0.5) | $\mathbf{99.94 \pm 0.1}$% | $\mathbf{30.43 \pm 0.3}$ % | $3.69 \pm 0.02$ % |
| GSP ($\alpha$=1.0) | $100.01 \pm 0.2$ % | $30.44 \pm 0.4$ % | $3.69 \pm 0.04$ % |
| Planned Shipments | $279.72$ % | $34.65 \pm 0.5$ % | $3.67 \pm 0.03$ % |
| Croston's Method | $1541.79$ % | $55.06 \pm 0.4$ % | $\mathbf{3.47 \pm 0.02}$ % |

Table 1: Prediction Performances (Mean ± SD, Calculated Across 4 Weeks)

Our experiments were performed using the historical data from a global consumer goods company with complex supply chain networks. The GSP models were trained on 18 months of historical data from March 2021 to August 2022, validated on the subsequent 4 months, and then tested on a 4-month hold-out dataset from 2023. The dataset covers 51 high-volume SKUs. Each SKU-week combination has a unique network graph topology. The networks include a varying number of nodes, from 2 to 50, and a varying number of edges, from 1 to 91.

In all our experiments with different methods, we consistently employed 4-week ahead demand predictions at the SKU/node level with wMAPEs ranging from 90% to 105% by forecasted timestep.

We also utilized the 4-week ahead short-term planned shipments ($|H| = 28, |W| = 4$) and a separately trained edge-level model for probabilistic lead time prediction, $P_{\text{LT},vw}^{t+h}(\text{k})$.[7] The GSP models were trained across all SKUs, each having individual quantity ranges as well as unique topological networks that may vary over time. All relevant variables, such as demand, supply, inventory are scaled to a uniform unit using a single SKU-specific scaler based on the maximum planned shipment quantity that occurred during the training data period. The GNN embedding layer, $\text{GAT}_{\text{XA}}$, was constructed using the PyTorch Geometric (PyG) library (Fey & Lenssen (2019); Paszke et al. (2019)), using GATv2Conv (Brody et al. (2022)). Using the validation dataset, we determined the optimal epoch and hyperparameters resulting in the minimum validation loss. More details on model training are available in Appendix B. We compared the prediction performances of GSP models against the planned shipments (original plan) and Croston's method on the testing dataset.

The planned shipments exhibited significant inaccuracies when compared to the actual shipments, and were frequently not executed as intended. Croston's method uses historical estimates of the average interval time between non-zero shipment events and the average non-zero shipment quantity for every edge, using smoothing parameters = 0.9. For GSP predictions, we use `hard = True` for `GumbelSoftmax`, enabling it to function as a categorical probability distribution. We predicted 4-week (28-day) horizon starting from each SKU/day network snapshot in the dataset, generating 20 probabilistic MC predictions. Table 1 compares different methods in terms of the performance metrics: (a) daily outgoing supply prediction sMACE, defined in Equation 1, (b) weekly inventory prediction wMAPE, defined in Equation 3, and (c) weekly constraint violation error $\kappa$ (i.e., a normalized metric indicating the extent to which the node-level weekly outgoing supply prediction quantities surpass the available supply capacity constraint, defined in Appendix E .

The results in Table 1 demonstrate that GSP models substantially outperformed the planned shipments and Croston's method in terms of both the edge-level daily outgoing supply sMACE and the node-level weekly inventory wMAPE. We also noted that Croston's method involved a substantial bias of 13.9%, whereas GSP ($\alpha = 0.5$) had a bias of 0.70% and planned shipments had a bias of 0.99%. In contrast, GSP ($\alpha = 0.5$, iteration = 0) showed minimal bias and yielded a $\kappa$ value that was comparable to that of the planned shipments.

Through our comprehensive investigations, we have been able to confirm that the outstanding performance metrics of GSP can mainly be attributed to its remarkable capability to detect systematic deviation patterns observed historically between actual and planned shipment events. These encompass deviations in both quantity (lower/higher) and timing (earlier/later) that are evident at the SKU/edge levels. For instance, certain nodes consistently delayed their outgoing supply events compared to the originally planned timings when there were recurring delays occurred in incoming supply from parent nodes. GSP effectively leveraged these patterns to enhance predictions for future event quantities and timings, all while adhering to constraints. We also noticed that GSP predictions based on GNN embeddings could incorporate the demand and inventory statuses of both source and destination nodes, as well as neighboring nodes. It is noteworthy that GSP ($\alpha = 0.5$), when equipped with a loss function that evenly weights errors in edge-level cumulative outgoing supply predictions and node-level inventory predictions, showcased the most exceptional overall performance.

# 7    CONCLUSION

Although accurate demand forecasting is a fundamental component of supply chain optimization, it is not the only necessity. Obtaining precise and reliable predictions for supply and inventory across all nodes and edges in supply chain networks is equally essential and challenging. We address this by GNN-based probabilistic approach that attains network-wide, reliable supply predictions while adhering to node-level supply capacity constraints. The designed loss function for model training, which combines cumulative supply prediction errors and inventory prediction errors, delivers robust performance on critical metrics. Our research plays a pivotal role in charting the path for the integration of AI within the supply chain domain.

---

[7]We think that the future extension of this paper may involve exploring the simultaneous training of models for both shipment events and lead time predictions.

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

## A    GRAPH ATTENTION NETWORKS (GAT)

We use Graph Attention Networks (GAT), a type of GNN architecture designed to assign dynamic attention weights to neighboring nodes in graph convolutions. In GAT, dynamic attentions enable a node to selectively attend to its neighboring nodes that are highly relevant in its supply predictions. Provided both node features $\mathbf{h}^{(0)}(=\mathbf{x})$ and edge features $\mathbf{e}\ (=\mathbf{a})$ for $\mathcal{G}=(\mathcal{V},\mathcal{E})$ with edges from neighboring source nodes $j \in \mathcal{N}_{\text{src}}(i) = \{j|(j,i) \in \mathcal{E}\}$ to node i, the update at layer $(l)$ is:

$$\mathrm{h}_i^{(l)} = \sigma(\alpha_{ii}^{(l-1)}\mathbf{W}_0^{(l-1)}\mathrm{h}_i^{(l-1)} + \sum_{j \in \mathcal{N}_{\text{src}}(i)} \alpha_{ij}^{(l-1)}\mathbf{W}_1^{(l-1)}\mathrm{h}_j^{(l-1)} + \mathbf{W}_2^{(l-1)}\mathrm{e}_{ji}) \qquad (18)$$

where $\alpha_{ij}^{(l-1)} = \text{softmax}_j(o_{ij}^{(l-1)}) \in \mathbb{R}$ is a dynamic attention[8] for edge $(j,i)$ using

$$o_{ij}^{(l-1)} = \mathbf{c}^{\mathsf{T}\,(l-1)}\,\text{LeakyReLU}\,(\mathbf{W}_0^{(l-1)}\mathrm{h}_i^{(l-1)} + \mathbf{W}_1^{(l-1)}\mathrm{h}_j^{(l-1)} + \mathbf{W}_2^{(l-1)}\mathrm{e}_{ji}). \qquad (19)$$

$\text{GAT}_{\text{XA}}$ denotes the $L$-layered graph convolution network that makes iterative updates $l = 1, 2, ..., L$ and calculates

$$\mathbf{h}^{(L)} = \text{GAT}_{\text{XA}}(\mathbf{h}^{(0)}, \mathbf{e}, \mathcal{G}; \phi_{\text{XA}}) = \{\mathrm{h}_v^{(L)} \in \mathbb{R}^B : v \in \mathcal{V}\} \in \mathbb{R}^{B \times |\mathcal{V}|} \qquad (20)$$

where $\phi_{\text{XA}} = \{\mathbf{W}_0^{(l-1)}, \mathbf{W}_1^{(l-1)}, \mathbf{W}_2^{(l-1)}, \mathbf{c}^{\mathsf{T}\,(l-1)} : l \in \{1, 2, ..., L\}\}$ are the set of all learned parameters of $\text{GAT}_{\text{XA}}$ and $B$ is the embedding dimension.

---

[8]This can be extended to multi-head attentions.

## B    Model Training Details

Using the Adam optimizer, we experimented with learning rates within the range of 1e-5 to 1e-2, incremented in factors of 10, varying the batch sizes among 1, 4, 8.

$GAT_{XA}$ incorporated LeakyReLU activation applied between each convolution layer. Using 2 or 3 convolution layers with 3 or 8 multi-head attentions for $GAT_{XA}$, we also varied the hidden dimension size. The $\texttt{mlp}_r$ employed 2 or 3 hidden layers with LeakyReLU activation before the output layer with a `sigmoid` output function. The $\texttt{mlp}_p$ employed 2 or 3 hidden layers with LeakyReLU activation before the output layer with a `GumbelSoftmax` output function.

Table 2 summarizes the selected optimal hyperparameter setting.

| Hyperparameter name | Selected value |
|---|---|
| epochs | 10 |
| learning rate | 0.0001 |
| batch size | 1 |
| # multi-head attentions | 3 |
| $GAT_{XA}$ graph convolution layers | [128, 32] |
| $\texttt{mlp}_r$ hidden layer size | [64, 32, 16] |
| $\texttt{mlp}_p$ hidden layer size | [15] |
| `GumbelSoftmax` temperature | 1.0 |

Table 2: The Best Hyperparameter Setting

## C    sMACE (scaled Mean Absolute Cumulative Error)

We define **sMACE (scaled Mean Absolute Cumulative Error)** as the mean of the absolute cumulative quantity errors (= samples of |cumulative predicted quantity - cumulative actual quantity|) divided by the mean of the actual quantities. Therefore, the daily outgoing supply prediction sMACE is defined as:

$$\textbf{sMACE} = \frac{\sum_{(t,(v,w))\sim\mathcal{D}}\sum_{h\in H}|\hat{Q}_{vw}^{\text{day},t}(h) - Q_{vw}^{\text{day},t}(h)|}{\sum_{(t,(v,w))\sim\mathcal{D}}\sum_{h\in H} q_{vw}^{\text{day},t}(h)} \times 100 \text{ \%} \tag{21}$$

where $\hat{Q}_{vw}^{\text{day},t}(h)$ and $Q_{vw}^{\text{day},t}(h)$ are the predicted and actual cumulative daily outgoing quantities from the forecasted timestep 0 to the forecasted timestep $h$ for the edge from node $v$ to node $w$ at the prediction time $t$ (= the start of day $t$), respectively. Also, $q_{vw}^{\text{day},t}(h)$ is the predicted daily outgoing quantity on the forecasted timestep $h$.

For the actual shipment quantity $a_{vw,\text{actual}}^{i|t}$ of event $i$ on the edge from source node $v$ to destination node $w$ at the prediction time $t$, if $a_{vw,\text{actual}}^{i|t}$ has been predicted to occure either $\delta_{vw,\text{actual}}^{i|t}$ timesteps earlier or later (i.e., predicted occurrance time $\hat{\tau}_{vw}^{i|t} = \tau_{vw,\text{actual}}^{i|t} + \delta_{vw,\text{actual}}^{i|t}$ ), the error contribution of $a_{vw,\text{actual}}^{i|t} \times |\delta_{vw,\text{actual}}^{i|t}|$ contributes to the numerator of sMACE.

For the sake of conceptual clarity, consider an illustrative scenario with actual quantities [0, 100, 0, 0] over 4 timesteps. The cumulative actual quantities = [0, 100, 100, 100]. We compare three predictive models, such as **M1** = [0, 0, 100, 0], **M2** = [100, 0, 0, 0], and **M3** = [0, 0, 0, 0].

M1 and M2 have a wMAPE of 200%, while M3 has 100%. Note that wMAPE (= weighted Mean Absolute Percentage Error) is defined as the sum of the absolute quantity errors (= samples of |predicted quantity - actual quantity|) divided by the sum of the actual quantities.

In the context of sMACE, when an actual quantity (representing a step increase in the cumulative actual quantity function) is predicted to occur either $d$ timesteps earlier ($d < 0$) or later ($d > 0$), it contributes an error equal to the quantity multiplied by $|d|$. Therefore, in terms of sMACE, both M1 and M2 have a score of 100%, whereas M3 scores 300%. In more detail,

- **M1**: 1-timestep late prediction for 100 actual quantity = $100 \times 1 = 100$ error;
  Cumulative predicted quantities = [0, 0, 100, 100];
  Absolute cumulative quantity errors = [0, 100, 0, 0];
  **sMACE** = 100/100 x 100% = 100%.

- **M2**: 1-timestep early prediction for 100 actual quantity = $100 \times 1 = 100$ error;
  Cumulative predicted quantities = [100, 100, 100, 100];
  Absolute cumulative quantity errors = [100, 0, 0, 0];
  **sMACE** = 100/100 x 100% = 100%.

- **M3**: at least 3-timestep late prediction for 100 actual quantity = $100 \times 3 = 300$ error;
  Cumulative predicted quantities = [0, 0, 0, 0];
  Absolute cumulative quantity errors = [0, 100, 100, 100];
  **sMACE** = 300/100 x 100% = 300%.

We may extend **sMACE** to involve a linear, increasing, or decreasing $\mathrm{penalty}$ function over the prediction time error of $\delta$, which is allowed to be defined in each domain of $\delta < 0$ and $\delta > 0$.

$$\mathrm{penalty}(\delta) = \begin{cases} \sum_{j=1}^{\delta} \varphi_+(j) & \text{if } \delta > 0 \\ \sum_{j=\delta}^{-1} \varphi_-(j) & \text{if } \delta < 0 \\ 0 & \text{if } \delta = 0 \end{cases} \tag{22}$$

where $\varphi_+(j)$ and $\varphi_-(j)$ are weight functions. Some possible choices for $\varphi_+(j)$ are defined as $c_1(1 + \eta_1(j-1))$ or $c_1\eta_2^{(j-1)}$. Similarly, $\varphi_-(j) = c_2(1 + \eta_3(|j| - 1))$ or $c_2\eta_4^{(|j|-1)}$. In these equations, $c_1$ and $c_2$ are positive constant cost parameters. Also, $\eta_1, \eta_3 \in [0, 1]$ are overweighting ratio parameters, whereas $\eta_2, \eta_4 \in (0, 1)$ are discounting ratio parameters.

$$\textbf{Generalized sMACE} = \frac{\sum_{(t,(v,w))\sim\mathcal{D}} \sum_{i\in\mathcal{A}_{vw,\text{actual}}^t} a_{vw,\text{actual}}^{i|t} \mathbb{E}_{\delta_{vw,\text{actual}}^{i|t}} \left[\mathrm{penalty}(\delta_{vw,\text{actual}}^{i|t})\right]}{\sum_{(t,(v,w))\sim\mathcal{D}} \sum_{i\in\mathcal{A}_{vw,\text{actual}}^t} a_{vw,\text{actual}}^{i|t}} \times 100\,\% \tag{23}$$

where $\mathcal{A}_{vw,\text{actual}}^t$ is the set of all actual events over the time horizon.

In this paper, we demonstrated the linear $\mathrm{penalty}(\delta) = |\delta|$, where $\varphi_+(j) = \varphi_-(j) = 1$.

**Generalized sMACE** can be associated with SPEC (Stock-keeping-oriented Prediction Error Cost) (Martin et al. (2020)) metric; however, it's important to note that SPEC is designed with the use of enforced weights, where $\varphi_+(j) = c_1 j$ and $\varphi_-(j) = c_2|j|$. This results in a quadratic penalty that grows with $|\delta|$, which may not be ideal for many situations. Also, it's worth mentioning that SPEC is not a normalized metric.

## D   INVENTORY PREDICTION PROCESS

We outline the execution steps in the inventory prediction process model $Z$.

Let $\mathcal{T}_{\text{week}}$ be the temporal transformation operator that aggregates daily samples into weekly bucketed samples, referring to the calendar information: Then, the predicted weekly quantity vector of outgoing supply is calculated as:

$$\hat{\boldsymbol{q}}_{vw}^{\text{week},t} = \mathcal{T}_{\text{week}}[\hat{\boldsymbol{q}}_{vw}^{\text{day},t}] \in \mathbb{R}^{|W|} \tag{24}$$

for the weekly time horizon $W = \{0, 1, 2, ..., |W| - 1\}$ where $|W| = |H|/7$. $\hat{q}_{vw}^{\text{day},t}(h)$ and $\hat{q}_{vw}^{\text{week},t}(\text{w})$ are the daily outgoing supply quantity at day $h \in H$ and the weekly outgoing supply quantity at week $\text{w} \in W$, respectively. This paper focuses on situations where weekly demand forecasting and planning features are accessible. Therefore, making weekly-granular inventory prediction is a reasonable choice.

Suppose that, for any outgoing supply at day $t + h$, the discrete probability distribution of the predicted lead time variable $\text{k} \in H$ is provided as $P_{\text{LT},vw}^{t+h}(\text{k})$ where $p_{\text{LT},vw}^{t+h}(k)$ is the probability

of predicted lead time k $= k$. The received (incoming) daily and weekly supply quantities at the destination node $w$ are computed as:

$$\hat{\boldsymbol{q}}_{vw,\text{recv}}^{\text{day},t} = \sum_{h \in H} \sum_{k \in H} \hat{q}_{vw}^{\text{day},t}(h) \, p_{\text{LT},vw}^{t+h}(k) \, \boldsymbol{e}^{(h+k)} \, , \quad \hat{\boldsymbol{q}}_{vw,\text{recv}}^{\text{week},t} = \mathcal{T}_{\text{week}}[\hat{\boldsymbol{q}}_{vw,\text{recv}}^{\text{day},t}] \tag{25}$$

$D_{v,\text{pred}}^{\text{w}|t}$, $\hat{S}_v^{\text{w}|t}$, and $\hat{A}_v^{\text{w}|t}$ are predicted demand, predicted incoming supply, and predicted outgoing supply for week w as predicted at the start of week $t$, respectively. Let $\mathcal{N}_{\text{dest}}(v)$ be the set of all neighboring destination (child) nodes $w$ for a given source node $v$. That is, $w \in \mathcal{N}_{\text{dest}}(v) = Pa_{\mathcal{G}^R}(v)$ where $\mathcal{G}^R$ is the reverse graph of $\mathcal{G}$ with the same nodes and features but with the directions of all edges reversed.

$$\text{predicted outgoing supply} \quad \hat{A}_v^{\text{w}|t} = \sum_{w \in \mathcal{N}_{\text{dest}}(v)} \hat{q}_{vw}^{\text{week},t}(\text{w}) \tag{26}$$

Let $\mathcal{N}_{\text{src}}(v)$ be the set of all neighboring source (parent) nodes $u'$ for a given node $v$. That is, $u' \in \mathcal{N}_{\text{src}}(v) = Pa_{\mathcal{G}}(v)$ where $\mathcal{G}$ is the network graph.

$$\text{predicted incoming supply} \quad \hat{S}_v^{\text{w}|t} = \sum_{u' \in \mathcal{N}_{\text{src}}(v)} \hat{q}_{u'v,\text{recv}}^{\text{week},t}(\text{w}) \tag{27}$$

It is crucial to satisfy that the predicted outgoing supply $\hat{A}_v^{\text{w}|t}$ should not be greater than the supply capacity constraint (i.e., no more outgoing supply than available capacity) $\hat{Y}_v^{\text{w}|t} = \hat{I}_v^{\text{w}|t} + \hat{S}_v^{\text{w}|t} - D_{v,\text{pred}}^{\text{w}|t}$. Thus, if $\hat{A}_v^{\text{w}|t} > \hat{Y}_v^{\text{w}|t}$, we make adjustments to $\hat{a}_{vw}^{i|t}$ for any event $i$ that impacts the outgoing supply at week w:

$$\hat{a}_{vw}^{i|t} := (\hat{Y}_v^{\text{w}|t} / \hat{A}_v^{\text{w}|t}) \, \hat{a}_{vw}^{i|t} \quad \text{for} \quad \text{any} \quad \text{event} \quad i \in \{i \in \mathbb{N} \mid u_{vw}^{i|t}(\text{w}) > 0\} \tag{28}$$

Then, the predicted inventory at the start of week w $+ 1$, $\hat{I}_v^{\text{w}+1|t}$ is updated as: for w $\in W$,

$$\text{predicted next start inventory} \quad \hat{I}_v^{\text{w}+1|t} = \hat{I}_v^{\text{w}|t} + \hat{S}_v^{\text{w}|t} - D_{v,\text{pred}}^{\text{w}|t} - \hat{A}_v^{\text{w}|t} \tag{29}$$

where $\hat{I}_v^{t|t}$ is the actual inventory at the start of week $t$, $I_{v,\text{actual}}^t$. Also, we define the weekly predicted inventory and weekly actual inventory vectors for node $v$ at week $t$ as:

$$\hat{\mathbf{I}}_v^{\text{week},t} = \{\hat{I}_v^{\text{w}|t} \mid \text{w} \in W\}, \qquad \mathbf{I}_v^{\text{week},t} = \{I_{v,\text{actual}}^{t+\text{w}} \mid \text{w} \in W\} \tag{30}$$

## E  SUPPLY CAPACITY CONSTRAINT VIOLATION ERROR

The weekly supply capacity constraint violation error $\kappa$ is defined as:

$$\kappa = \frac{\sum_{(t,v) \sim \mathcal{D}} \Sigma_{\{\text{w} \mid \hat{A}_v^{\text{w}|t} > \hat{Y}_v^{\text{w}|t}\}} (\hat{A}_v^{\text{w}|t} - \hat{Y}_v^{\text{w}|t})}{\sum_{(t,v) \sim \mathcal{D}} \Sigma_{\text{w} \in W} I_v^{\text{w}|t}} \times 100 \, \% \tag{31}$$

## F  ITERATIVE PREDICTIONS TO SATISFY THE SUPPLY CAPACITY CONSTRAINT

Suppose that any node in the supply chain network can use the supply it receives during a given week to fulfill both demand and outgoing supply needs for that same week. In this context, the incoming supply to a node, which includes shipments sent from its source nodes and received to that node within the same week, directly affects its supply capacity constraint to determine the maximum available outgoing supply during the same week. However, typical GNN implementations do not provide support for sequential supply predictions along cascading directional graphs; instead, they predict the supply quantities of all edges across the network simultaneously. As a result, we present an iterative inference approach to ensure that supply capacity constraints at each node are satisfied.

We use $\hat{\boldsymbol{q}}_{u'v}^{\text{day},t,\text{Iter}=0}$ to represent the initial unconstrained prediction of daily supply quantities on the edge from source node $u'$ and destination node $v$. Also, $\hat{\boldsymbol{q}}_{u'v,0:(7\text{w}-1)}^{\text{day},t,\text{Iter}=n}$ denotes the constrained prediction of daily supply quantities during the $n$-th iteration up to the start of week w ($= 0, 1, ..., |W|-1$)

| Iterations | (a) Daily Outgoing Supply sMACE | (b) Weekly Inventory wMAPE | (c) Weekly Constraint Error $\kappa$ |
|---|---|---|---|
| 0 | $99.94 \pm 0.1$ % | $30.43 \pm 0.3$ % | $3.69 \pm 0.02$ % |
| 1 | $99.89 \pm 0.2$ % | $30.40 \pm 0.3$ % | $3.57 \pm 0.02$ % |
| 2 ($\rho < 0.005$) | $\mathbf{98.58 \pm 0.1}$ % | $\mathbf{29.92 \pm 0.2}$ % | $\mathbf{3.45 \pm 0.01}$ % |

Table 3: GSP ($\alpha$=0.5) Performances for Iterations (Mean $\pm$ SD, Calculated Across 4 Weeks)

based on the predictions from the previous $(n-1)$-th iteration. Furthermore, $\hat{\boldsymbol{q}}_{u'v,\ 7\mathrm{w}:(7|W|-1)}^{\mathrm{day},t,\mathrm{Iter}=(n-1)}$ is the prediction made during the $(n-1)$-th iteration, spanning from week w to week $(|W|-1)$. Then, we define the concatenated vector that represents the revised prediction for the entire time horizon $(7|W| = |H|$ days) estimated at the start of week w during the $n$-th iteration:

$$\hat{\boldsymbol{q}}_{u'v,\ \mathrm{est\ at\ w}}^{\mathrm{day},t,\mathrm{Iter}=n} = \hat{\boldsymbol{q}}_{u'v,0:(7\mathrm{w}-1)}^{\mathrm{day},t,\mathrm{Iter}=n} \oplus \hat{\boldsymbol{q}}_{u'v,\ 7\mathrm{w}:(7|W|-1)}^{\mathrm{day},t,\mathrm{Iter}=(n-1)} \tag{32}$$

where $\oplus$ is the vector concatenation operator. Next, the received (incoming) daily and weekly supply quantity vectors at the node $v$ in any iteration $n \geq 1$ is computed as:

$$\hat{\boldsymbol{q}}_{u'v,\ \mathrm{est\ at\ w,\ recv}}^{\mathrm{day},t,\mathrm{Iter}=n} = \sum_{h \in H} \sum_{k \in H} \hat{\boldsymbol{q}}_{u'v,\ \mathrm{est\ at\ w}}^{\mathrm{day},t,\mathrm{Iter}=n}(h)\ p_{\mathrm{LT},vw}^{t+h}(k)\ \boldsymbol{e}^{(h+k)} \tag{33}$$

$$\hat{\boldsymbol{q}}_{u'v,\ \mathrm{est\ at\ w,\ recv}}^{\mathrm{week},t,\mathrm{Iter}=n} = \mathcal{T}_{\mathrm{week}}[\hat{\boldsymbol{q}}_{u'v,\ \mathrm{est\ at\ w,\ recv}}^{\mathrm{day},t,\mathrm{Iter}=n}] \tag{34}$$

In each iteration $n(\geq 1)$, for every week $w = 0, 1, ..., |W| - 1$, we update the node $v$'s incoming supply $\hat{S}_v^{\mathrm{w}|t}$ and outgoing supply capacity $\hat{Y}_v^{\mathrm{w}|t}$.

$$\hat{S}_v^{\mathrm{w}|t} = \sum_{u' \in \mathcal{N}_{\mathrm{src}}(v)} \hat{q}_{u'v,\ \mathrm{est\ at\ w,\ recv}}^{\mathrm{week},t,\mathrm{Iter}=n}, \qquad \hat{Y}_v^{\mathrm{w}|t} = \hat{I}_v^{\mathrm{w}|t} + \hat{S}_v^{\mathrm{w}|t} - D_{v,\mathrm{pred}}^{\mathrm{w}|t} \tag{35}$$

Then, if the predicted outgoing supply $\hat{A}_v^{\mathrm{w}|t} = \sum_{w \in \mathcal{N}_{\mathrm{dest}}(v)} \hat{q}_{vw}^{\mathrm{week},t}(\mathrm{w})$ is greater than the supply capacity constraint $\hat{Y}_v^{\mathrm{w}|t}$, we make adjustments to $\hat{a}_{vw}^{i|t}$ for any event $i$ that impacts the outgoing supply at week w:

$$\mathrm{if}\ \ \hat{A}_v^{\mathrm{w}|t} > \hat{Y}_v^{\mathrm{w}|t}, \quad \hat{a}_{vw}^{i|t} := (\hat{Y}_v^{\mathrm{w}|t} / \hat{A}_v^{\mathrm{w}|t})\ \hat{a}_{vw}^{i|t} \quad \mathrm{for\ any\ \ event}\ \ i \in \{i \in \mathbb{N} \mid u_{vw}^{i|t}(\mathrm{w}) > 0\} \tag{36}$$

Accordingly, we update $\hat{\boldsymbol{q}}_{vw}^{\mathrm{day},t,\mathrm{Iter}=n}$ and next start inventory $\hat{I}_v^{\mathrm{w}+1|t} = \hat{I}_v^{\mathrm{w}|t} + \hat{S}_v^{\mathrm{w}|t} - D_{v,\mathrm{pred}}^{\mathrm{w}|t} - \hat{A}_v^{\mathrm{w}|t}$.

We do multiple iterations until the averaged relative change $\rho$ becomes smaller than $\epsilon$.

$$\rho = \mathbb{E}_{(t,(u',v)) \sim \mathcal{D}} \left[ \frac{\| \hat{\boldsymbol{q}}_{u'v}^{\mathrm{day},t,\mathrm{Iter}=n} - \hat{\boldsymbol{q}}_{u'v}^{\mathrm{day},t,\mathrm{Iter}=n-1} \|_2}{\| \hat{\boldsymbol{q}}_{u'v}^{\mathrm{day},t,\mathrm{Iter}=n-1} \|_2} \right] < \epsilon \tag{37}$$

where $\epsilon$ is a small positive threshold hyperparameter (e.g., 0.005).

In Table 3, we also presented how the use of iterative inferences contributes to the improvement of performance metrics, notably in reducing the constraint-violation error $\kappa$. This involved a comparison between scenarios with no iterations (unconstrained prediction) and varying numbers of iterations until convergence.

