# OpenReview forum: "GNN-based Probabilistic Supply and Inventory Predictions in Supply Chain Networks"
_ICLR.cc/2024/Conference — Submitted to ICLR 2024_

### Official Review · Reviewer_4JxK · 2023-10-31

**Soundness:** 2 fair
**Presentation:** 2 fair
**Contribution:** 1 poor
**Rating:** 3
**Confidence:** 4

**Summary:**

This paper proposes a predictor for the lane-level cumulative outgoing supply along with the node-level inventory levels. The solution is based on a Graph attention mechanism. The authors show that their predictor performs well on a real dataset.

**Strengths:**

The method seems to solve the problem for the industry sponsor of the project.

**Weaknesses:**

1. The authors do not state the contribution of the paper. I am having a hard time judging the significance of the problem and the results.
2. Numerical experiments lack competing baselines. The numbers in Table 1 and 2 thus do not carry a lot of meanings. The rationale behind incorporating demand prediction inputs in the inventory prediction is not clearly stated. The numerical results only compare the performance when the hyperparameter is chosen from {0,0.5,1}.
3. What is a lane? Is it an edge on the graph? This notion is quite untraditional both in the supply chain literature and the graph learning literature.
4. There are no innovations in terms of neural network architectures.
5. The design of the solution seems ad-hoc. It is unclear how the solution package proposed in this paper can be generalized to solve other practical problems. The paper also lacks some theoretical contributions to justify the soundness of the approach.


As I am also a researcher working on the intersection of machine learning and supply chain management, I do not see that this paper fits with the audience of this conference. Simply using a GAN to solve a specific dataset does not excite the audience of this community.

**Questions:**

1. I find the discussion in Section 3.2 intimidating with an overload of indices. To my understanding, the difficulty of this paper only lies in the construction of the target variables (partly due to the dynamism of the system). Section 3.2 and 3.3 are simply preparing these target values, so I think the authors should make this information simpler. So far, Section 3.2 and 3.3 look more like a "project report" to the industry partner. For the audience of this conference, why should we care about these calculations leading to equation (12) and (18)? In the end, neural networks are just functional approximations: we match input with output. Thus, even though the output can be difficult to compute, preparing the target variables is just data-processing. Unfortunately, it may not be exciting for the audience of the ICLR conference to learn about how the authors process the data.


// Post-rebuttal: I thank the authors for their replies. I slightly increase my score.

---

> ### Author Response · Authors · 2023-11-17
>
> Thank you for your valuable feedback. We have made significant revisions to enhance the clarity and readability of our paper. In the revised version,
>
> - Added explicit descriptions of our main contributions
>
> - Provided a clear description (Section 3) of the Problem Description.
>
> - Introduced our generalized method (Section 4) for GNN-based event quantity/timing delta predictions
>
> - Presented our GSP (GNN-based Supply Prediction) models as an application of the generalized method for shipment event predictions in supply chain networks (Section 5)
>
> - Addressed the clarity problems you highlighted.
>
> - Consistently use the term “edge” instead of “lane”
>
> ---
> 1. Generalizability, Problem Setting, Novelty
>
> Our method is very generalizable for making event quantity/timing delta predictions in graph-structured problem contexts where there are planned events without a one-to-one mapping to actual events. This implies that there is no ground truth as labeled data for quantity/timing variables for model learning. Our approach relies on labeled data in the form of edge-level and/or node-level aggregated quantities at specific temporal granularities, such as daily or weekly periods.
>
> Our revised paper now clearly introduces the generalized method (Section 4) for GNN-based event quantity/timing delta predictions. Subsequently, in Section 5, we present our GSP (GNN-based Supply Prediction) models as an application of this generalized method.
>
> Our method is versatile and can be applied to more general cases of event delta predictions in graph-structured settings, given some known prior or planned event information. For instance, it can be employed in any time series forecasting with intermittent event data where the prediction objective is centered around cumulative quantity prediction accuracy over a specific time horizon, provided some historically estimated prior information of events.
>
> In designing our loss function, we provided flexibility for the model to utilize either the edge-level aggregated quantity labels (e.g., daily supply on edges) or the node-level aggregated quantity labels (e.g., weekly inventory on nodes), or a combination of both.
>
> ---
>
> 2. Inventory Prediction Process, Backprop through the Process
>
> In response to your feedback, in our revised paper, Equation (15) for the generalized method introduces a pre-determined process model Z that takes edge-level predicted quantity vectors and predicts node-level aggregated quantity vectors.  This process model corresponds to the inventory prediction process in supply chain contexts.  We moved the details regarding the inventory prediction process to Appendix D.
>
> Still, we’d like to highlight that, due to limitations in the underlying supply chain processes and systems for tracking actual shipments against planned shipments, it is often not possible to establish a direct one-to-one correspondence between actual and planned shipments. Consequently, we lack precise information regarding the exact discrepancies between actual and planned shipments in terms of both shipment event quantity and timing. To address this issue, GSP model learns from the accessible labels (ground truth) in the form of the edge-level daily outgoing supply quantities and the node-level weekly inventory quantities.
>
> Hence, we contend that this model goes beyond a standard GAN/neural network designed to match inputs with outputs. The uniqueness lies in how the labels across the graph are provided, and the calculation of the loss function is executed. We compute cumulative supply prediction loss to encompass errors in both event quantity and timing simultaneously, predict (or reconstruct) inventory within the model, and employ backpropagation through the inventory prediction process model. This approach enables the model to leverage both edge-level and node-level aggregated labels at varying temporal granularities. Our approach can be also successfully employed for predictions in scenarios where events are intermittent, errors may accumulate, and graphically-aggregated labels exist at different temporal granularities.
>
> ---
>
> 3. ICLR Relevance
>
> We think that our generalized method provides the solution to a very interesting “supervised representation learning” problem (ICLR’s one of main topics) in which the labels (ground truth) are not directly provided in the form of event quantity/timing (main prediction targets), but only obtainable in the form of edge-level and/or node-level aggregated quantities at specific temporal granularities.
> The ICLR website indicates that one of the relevant topics is “applications in audio, speech, robotics, neuroscience,  biology, or any other field.”  We believe that our research may play a pivotal role in charting the path for the integration of AI within the supply chain domain.

---

> > ### Comment · Reviewer_4JxK · 2023-11-22
> > **Clarification on representation learning**
> >
> > // Sorry for this late reply.
> >
> > I am confused now because you say the paper is about representation learning. I have read through the revised paper (thanks for updating), but I do not see the representation learning flavors in this paper. If you have time, could you please elaborate? [I am looking at equation (16)  as the main contribution of this paper, but it is more a regression, prediction problem than a representation learning problem]

---

> ### Author Response · Authors · 2023-11-22
> **graph node embedding representation**
>
> We are grateful for your time in reviewing our revised paper!  Hope our response positively impacts your support of accepting our paper.
>
> We'd like to underscore that our GNN (Graph Attention Network - GAT) layers, as defined in Equation (5), are dedicated to learning embedding representations (graph node embeddings) to accomplish predictive learning tasks. Notably, our approach avoids intricate and manual feature engineering tasks. The loss function we have formulated is tailored for GNN to optimize embedding representations, minimizing prediction losses for both edge-level and node-level labels. Our subsequent analysis, employing the UMAP visualization technique on Graph Node Embeddings, reveals that the GNN effectively learns representations and discerns patterns. We hope that you will find this response relevant.

---

### Official Review · Reviewer_eXox · 2023-10-31

**Soundness:** 2 fair
**Presentation:** 1 poor
**Contribution:** 1 poor
**Rating:** 3
**Confidence:** 3

**Summary:**

The paper proposes to address the problem of predicting supply and inventory in supply chain networks using a GNN-based method.

**Strengths:**

*Originality*
The work demonstrates the capability of AI adoption in a relatively uncommon domain. The author designed new lost function and iterative procedure for ensuring feasibility.

*Quality and Clarity*
The author introduces clearly the motivation of the research. The authors made successful use of figures and tables that helped understanding the background and methodology.

*Significance*
The authors are able to demonstrate the efficacy of their method in predicting supply for a historical data.

**Weaknesses:**

My major concerns are with the presentation and clarity. It is almost impossible to follow the details and check the correctness of the methods introduced. The authors directly dive into the methodology without clearly state the problem, and combined the modeling with discussions of GAT. It is rarely possible to immediately distinguish which parameters are known before making the prediction, which are decision variables. What are inputs of your prediction? What is the definition of planned shipments? Do you know a plan of future supply before making prediction? Not mentioning that the overwhelming number of parameters and notations used without sufficient explanations make it even tougher.  It is a must to explicitly setup the problem and introduce all assumptions and formulation details of your supply chain model, especially when it seems that your formulation and model of the supply chain is different from traditional literature.

Other minor presentation problems include:
- It is uncommon in the network literature using the term "lane"; commonly used are edges or arcs. The author used "edges" when describing the graph but used "lane" elsewhere. It is misleading.
- Overuse of footnotes is annoying. If the texts are important, bring them to the main text; if not, delete it or put in the supplementals.
- The notations are bad and unreadable. For example, 7 super/sub-scripts for a single variable
$$\hat{q}^{day, t, Iter=n}_{u, v, est at w, recv}$$

*Contributions*

I may misunderstand some parts (for reasons stated above) but I personally don't think this paper makes sufficient realistic contribution to the real supply chain planning problems. From a macroscopic view, the prediction problem makes plausible sense and the ability to predict the incoming of supply is good. In reality, the supply chains are rather decentralized and divided without a centralized decision maker that overlook everything and make the planning.  Local decision makers meet much smaller-scale problems and have more well-controlled supplier relationships that leads to accuracte local prediction of incoming supply. Collecting data and make centralized prediction can be infeasible and unnecessary.

The authors claim contributions on the design of loss function, which is bi-objective controlled by a weight hyperparameter. In optimization society, it is usually wise to avoid such formulation because it is hard to interpret and justify the results, especially when your two objectives are in different scale (one daily error, and the other weekly cumulated errors). How can you quantify the impacts of the two objectives to the followup planning? Which error will result in higher operational costs?

**Questions:**

- The baseline methods are obviously not good enough and compare your method with them does not stand you out. Are they the only options people can use to predict the ongoing supply?

- Is the method introduced robust to disruptions? It is one of the most important questions to think about when claiming the usefulness. If there is SKU shutting down or suddenly high labor shortage (as in the pandemic), I doubt if the method can quickly capture the dynamics and still have good performance.

---

> ### Author Response · Authors · 2023-11-17
>
> Thank you for your valuable feedback. We have made significant revisions to enhance the clarity and readability of our paper. In the revised version,
>
> - Provided a clear description (Section 3) of the Problem Description.
>
> - Clearly indicated the target variables for prediction, the labeled variables for model learning, and the input feature variables.
>
> - Introduced our generalized method (Section 4) for GNN-based event quantity/timing delta predictions
>
> - Presented our GSP (GNN-based Supply Prediction) models as an application of the generalized method for shipment event predictions in supply chain networks (Section 5)
>
> - Addressed the presentation problems you highlighted.
>
> ---
>
> 1. Generalizability, Problem Setting, Novelty
>
> Our method is very generalizable for making event quantity/timing delta predictions in graph-structured problem contexts where there are planned events without a one-to-one mapping to actual events. This implies that there is no ground truth as labeled data for quantity/timing variables for model learning. Our approach relies on labeled data in the form of edge-level and/or node-level aggregated quantities at specific temporal granularities, such as daily or weekly periods.
>
> Our revised paper now clearly introduces the generalized method (Section 4) for GNN-based event quantity/timing delta predictions. Subsequently, in Section 5, we present our GSP (GNN-based Supply Prediction) models as an application of this generalized method.
>
> Our method is versatile and can be applied to more general cases of event delta predictions in graph-structured settings, given some known prior or planned event information. For instance, it can be employed in any time series forecasting with intermittent event data where the prediction objective is centered around cumulative quantity prediction accuracy over a specific time horizon, provided some historically estimated prior information of events.
>
> ---
>
> 2. Planned Shipments
>
> Planned shipments mean the original scheduled (planned) shipment events in the enterprise’s planning system. The planned shipments frequently fail to be executed as originally devised. We use the planned shipment as prior information when predicting the deltas (= differences between the plan and the actual) in quantity/timing.
>
> ---
>
> 3. Usefulness for Real-World Supply Chain Systems
>
> We respectfully disagree with your point of view that collecting data and making centralized predictions can be infeasible and unnecessary in real-world supply chain systems. Many organizations employ established S&OP processes for centralized predictions and planning across nodes and lanes in the supply chain network. Therefore, accurately predicting supply executions in relation to the original plan is crucial for organizations to make better informed corrections to their existing plans. We believe that our work may play a pivotal role in charting the path for the integration of AI within the supply chain domain.
>
> ---
>
> 4. Baselines
>
> In our paper, we have included two primary baselines—the error of the original plan and the Croston method's error.
>
> As highlighted in the Related Work section, our problem setting significantly diverges from those of traditional approaches. There are currently no open-source dataset and state-of-the-art method/code available for assessing supply prediction performance in topologically complex supply chain networks at the scale we addressed in this paper. We aim for our work to establish a foundational reference, encouraging others to focus on AI applications within the realm of supply chain networks.
>
> ---
>
> 5. Loss Function Design
>
> In designing our loss function, we provided flexibility for the model to utilize either the edge-level aggregated quantity labels (e.g., daily supply on edges) or the node-level aggregated quantity labels (e.g., weekly inventory on nodes), or a combination of both. While we acknowledge your observation, we believe that employing a weight hyperparameter to balance different loss components is a common practice, often used for regularization techniques. Additionally, optimizing the weight hyperparameter to achieve a balance between the two prediction objectives poses a similar challenge to tuning other model hyperparameters, as demonstrated in our experiment results.
>
> ---
>
> 6. Supply Chain Disruption, Robustness
>
> Our work exhibits robustness in the face of supply chain disruptions, including sudden changes in network topology. This resilience stems from our models learning from a vast amount of topologically diverse structured data during the training. Even when certain nodes or lanes are removed from the network topology at the time of inference, the model continues to identify patterns in the new topology and makes good predictions, as observed previously.

---

### Official Review · Reviewer_67xi · 2023-11-01

**Soundness:** 3 good
**Presentation:** 3 good
**Contribution:** 2 fair
**Rating:** 5
**Confidence:** 5

**Summary:**

The authors propose a novel GNN-based architecture to predict supplies, inventory, and imbalances for supply chain optimization. This is achieved by introducing (i) a probabilistic model that operates on carefully designed representations (e.g., discretized timing-deltas for an event or multipliers for event quantities), and (ii) an iterative inference approach to ensure supply capacity constraints.

Empirically, the authors show that the proposed method outperforms domain-specific algorithms (i.e., Croston’s method) and planned shipments from the dataset using historical data from a global consumer goods company.

**Strengths:**

I am not an expert in the field of supply chain forecasting, but I enjoyed the thought process behind the development of the architecture. The authors encode specific domain knowledge to achieve good performance and reliability (e.g., constraint satisfaction).

Despite the readability of Section 3 could be improved, the narration is fairly clear and complete. Figures are also generally informative and overall the paper is well written.

**Weaknesses:**

In my opinion, the main weaknesses of this work lie in (i) the extreme specificity of the methodology, (ii) the lack of baselines and/or experiments to validate the proposed architectural innovations, and (iii) more generally, the relevance to the broader ICLR community.

**Questions:**

(i) The extreme specificity of the methodology:
Supply chain is an extremely relevant problem. However, the proposed architecture seems to be extremely tailored for this one specific application. How generalizable are these methods beyond the supply chain application? It'd be nice to see experiments on a more diverse set of problems.


(ii) The lack of baselines and/or experiments to validate the proposed architectural innovations:
Arguably, the major contribution of this work is the neural network architecture and input/output representations. The authors do a good job at motivating (in text) the reasoning behind their choices, however, there is little to no evidence to support the specific choices. Can the authors provide ablations in support of the individual choices in the architecture? For example:
- Discrete time-deltas vs continuous representation
- Delta vs Non-delta
- Quantity multiplier vs absolute number
- etc.

Moreover, in the same direction, the set of baselines is extremely limited. The authors should provide additional learning-based prediction models from literature or simply by implementing sensible alternative approaches to the problem

---

> ### Author Response · Authors · 2023-11-17
>
> Thanks for your valuable feedback. We hope our response addresses your questions. Please note that we posted our revised paper.
>
> ---
>
> 1. Generalizability, Problem Setting, Novelty
>
> Our method is very generalizable for making event quantity/timing delta predictions in graph-structured problem contexts where there are planned events without a one-to-one mapping to actual events. This implies that there is no ground truth as labeled data for quantity/timing variables for model learning. Our approach relies on labeled data in the form of edge-level and/or node-level aggregated quantities at specific temporal granularities, such as daily or weekly periods.
>
> Our revised paper now clearly introduces the generalized method (Section 4) for GNN-based event quantity/timing delta predictions. Subsequently, in Section 5, we present our GSP (GNN-based Supply Prediction) models as an application of this generalized method.
>
> Our method is versatile and can be applied to more general cases of event delta predictions in graph-structured settings, given some known prior or planned event information. For instance, it can be employed in any time series forecasting with intermittent event data where the prediction objective is centered around cumulative quantity prediction accuracy over a specific time horizon, provided some historically estimated prior information of events.
>
> In designing our loss function, we provided flexibility for the model to utilize either the edge-level aggregated quantity labels (e.g., daily supply on edges) or the node-level aggregated quantity labels (e.g., weekly inventory on nodes), or a combination of both.
>
> ---
>
> 2. Baselines
>
> Thank you for your insightful suggestions regarding more interesting baselines. We value your input!
>
> In our paper, we have included two primary baselines—the error of the original plan and the Croston method's error.
>
> As highlighted in the Related Work section, our problem setting significantly diverges from those of traditional approaches. There are currently no open-source dataset and state-of-the-art method/code available for assessing supply prediction performance in topologically complex supply chain networks at the scale we addressed in this paper. We aim for our work to establish a foundational reference, encouraging others to focus on AI applications within the realm of supply chain networks.
>
> ---
>
> 3. ICLR Relevance
>
> We think that our generalized method provides the solution to a very interesting “supervised representation learning” problem (ICLR’s one of main topics) in which the labels (ground truth) are not directly provided in the form of event quantity/timing (main prediction targets), but only obtainable in the form of edge-level and/or node-level aggregated quantities at specific temporal granularities.
>
> The ICLR website indicates that one of the relevant topics is “applications in audio, speech, robotics, neuroscience,  biology, or any other field.”  We believe that our research may play a pivotal role in charting the path for the integration of AI within the supply chain domain.

---

### Official Review · Reviewer_vuxs · 2023-11-10

**Soundness:** 3 good
**Presentation:** 3 good
**Contribution:** 2 fair
**Rating:** 6
**Confidence:** 3

**Summary:**

This article presents the highly researched problem i.e. supply and demand imbalances over time using Graph-based supply prediction probabilistic model that predicts supplies, inventory, and imbalances using graph-structured historical data, demand forecasting, and original supply plan puts.

**Strengths:**

Good detailing of the problem statement and to point literature review to distinguish the contribution from the traditional prediction scenarios.

**Weaknesses:**

One of the key problems in supply chain networks is lead time predictions and probably a potential extension the authors can think about extending it.

**Questions:**

GATs apply stacked layers in which nodes consists feature of neighboring nodes, applying these attention makes the whole network with different weights to the different nodes present in the neighbors which was leveraged to predict supply & that's interesting. The authors need to explore a bit more in detail in section 3.1 for better readability.

Is there any stabilization process carried out in output layer such as multi-head attention?
any sort of transformation or concatenation is done on the output features?


Typo error could have been avoided in Page 4 -foot note no. 3 - dailiy basis needs attention.

---

> ### Author Response · Authors · 2023-11-17
>
> Thanks for your valuable feedback.  We hope our response addresses your questions. Please note that we have made significant revisions to enhance the clarity and readability of our paper.
>
> 1.	We envision that future extensions of this paper may involve the simultaneous training of models for both shipment events and lead time predictions. We mentioned this in footnote 7 in our paper.
> 2.	In our experiments we performed hyperparmeter tuning as described in Appendix B (Model Training Details). We experimented with 2 or 3 GAT convolution layers with 3 or 8 multi-head attentions using the validation dataset.  We determined 2 GAT convolution layers with 3 multi-head attentions that result in the minimum validation loss.
> 3.	The transformation is done to the model outputs in the format as shown in Figure 8. All values are normalized using a min-max normalization, where the min value is normally zero.
> 4.	We corrected typos in our revised paper.
>
> Moreover, we'd like to emphasize that our method is very generalizable for making event quantity/timing delta predictions in graph-structured problem contexts where there are planned events without a one-to-one mapping to actual events. This implies that there is no ground truth as labeled data for quantity/timing variables for model learning. Our approach relies on labeled data in the form of edge-level and/or node-level aggregated quantities at specific temporal granularities, such as daily or weekly periods.
>
> Our revised paper now clearly introduces the generalized method (Section 4) for GNN-based event quantity/timing delta predictions. Subsequently, in Section 5, we present our GSP (GNN-based Supply Prediction) models as an application of this generalized method.
>
> Our method is versatile and can be applied to more general cases of event delta predictions in graph-structured settings, given some known prior or planned event information. For instance, it can be employed in any time series forecasting with intermittent event data where the prediction objective is centered around cumulative quantity prediction accuracy over a specific time horizon, provided some historically estimated prior information of events.
>
> In designing our loss function, we provided flexibility for the model to utilize either the edge-level aggregated quantity labels (e.g., daily supply on edges) or the node-level aggregated quantity labels (e.g., weekly inventory on nodes), or a combination of both.

---

### Author Response · Authors · 2023-11-17
**Revised paper posted**

We appreciate your feedback and hope our response positively impacts your support of accepting our paper.

Based on your constructive reviews, we implemented a few significant changes to enhance our paper:

- Restructured the paper for improved clarity and readability
- Added explicit descriptions of our main contributions
- Provided a clear description (Section 3) of the Problem Description.
- Introduced our generalized method (Section 4) for GNN-based event quantity/timing delta predictions
- Presented our GSP (GNN-based Supply Prediction) models as an application of the generalized method for shipment event predictions in supply chain networks (Section 5)

We hope you will have the opportunity to review our updated paper, which incorporates your main feedback.

---

### Meta-Review · Area_Chair_Nbmx · 2023-12-08

**Metareview:**

The paper proposes a graph NN method for supply chain predictions, and compares its efficacy vs. existing heuristics using historical data from a large-scale real-world supply chain. The reviewers acknowledged the interesting insights in state and graph-representation for the problem, but had concerns about the generality of the approach beyond a very narrow domain. During the rebuttal period, the authors added a section to the paper describing the general problem of making event-level predictions but only observing aggregated feedback. The paper will be substantially stronger if there are experimental results of the proposed GNN approach in other domains where the general problem arises, thereby demonstrating the applicability of the method across a wider array of applications.

**Justification For Why Not Higher Score:**

All the reviewers raised concerns about the generalizability of the proposed method for problems beyond the single application that the authors tested on in the paper. The authors included a good characterization of the class of problems that the method may be suitable for (beyond the one application that they tested on). Including examples of this class of problems, and empirically benchmarking the approach along with ML baselines will substantially strengthen the paper. Additionally, conducting ablations to pin-point the benefits of different design choices will help other practitioners identify how to apply this work to their problems.

**Justification For Why Not Lower Score:**

N/A

---

### Decision · Program_Chairs · 2024-01-16

Reject